# LM vs LM: Detecting Factual Errors via Cross Examination

**Roi Cohen**[1]   **May Hamri**[1]   **Mor Geva**[2]   **Amir Globerson**[1,3]

[1]Blavatnik School of Computer Science, Tel Aviv University

[2]Google DeepMind   [3]Google Research

{roi1, mayhamri}@mail.tau.ac.il, pipek@google.com, gamir@tauex.tau.ac.il

## Abstract

A prominent weakness of modern language models (LMs) is their tendency to generate factually incorrect text, which hinders their usability. A natural question is whether such factual errors can be detected automatically. Inspired by truth-seeking mechanisms in law, we propose a factuality evaluation framework for LMs that is based on cross-examination. Our key idea is that an incorrect claim is likely to result in inconsistency with other claims that the model generates. To discover such inconsistencies, we facilitate a multi-turn interaction between the LM that generated the claim and another LM (acting as an examiner) which introduces questions to discover inconsistencies. We empirically evaluate our method on factual claims made by multiple recent LMs on four benchmarks, finding that it outperforms existing methods and baselines, often by a large gap. Our results demonstrate the potential of using interacting LMs to capture factual errors.

## 1 Introduction

Modern language models (LMs) often generate inconsistent (Elazar et al., 2021), non-attributable (Rashkin et al., 2021; Bohnet et al., 2022; Liu et al., 2023a), or factually incorrect text (Tam et al., 2022; Devaraj et al., 2022; Maynez et al., 2020), thus negatively impacting the reliability of these models (Amodei et al., 2016; Hendrycks et al., 2021). This has prompted the community to develop methods that calibrate the confidence of model predictions to better align with their quality (Brundage et al., 2020). For example, prior methods have used probabilistic approaches (Jiang et al., 2020; Zablotskaia et al., 2023) clustering (Kuhn et al., 2023), fine-tuning (Kadavath et al., 2022; Lin et al., 2022) and in-context learning (Alivanistos et al., 2022; Cohen et al., 2023).

In this work, we take a different approach to this problem, motivated by truth-seeking mechanisms in law. Specifically, we consider the setting where a witness is cross-examined in order to check whether their statement is factually correct or not. In such a setting, the examiner asks questions that aim to lead towards contradictory statements by the witness, and a contradiction implies that the witness lied at least in some of the statements. Hence the well known quote *"Were you lying then or are you lying now?"* (Wilder et al., 1957).

To employ this mechanism to LM factual calibration, we propose the following setting, illustrated in Figure 1. Our goal is to check whether a statement made by an LM (*"Augustus was the first Roman Emperor to sport a beard"*) is factually correct. We refer to the model that generated this statement as the EXAMINEE. To check whether this fact is correct, we use another LM, called EXAMINER, to conduct a cross-examination of EXAMINEE. Concretely, we craft designated prompts to facilitate a multi-turn interaction between the two LMs, where EXAMINER issues questions (e.g., *"Was it common for Roman Emperors to wear beards?"*) to EXAMINEE to check the veracity of the original statement. The examination is concluded by a decision from EXAMINER as to whether the original claim was correct or not.[1]

Our problem setting is related to that of calibration (Guo et al., 2017), where the goal is to predict the probability at which a model will err. However, unlike previous approaches to this problem, we use text generated by LMs. Our approach is motivated by the intuition that calibration is actually an elaborate reasoning process where one checks the level of support that a fact has, based on other statements the model believes. We argue that such complex reasoning is naturally performed via the strong conversational skills of modern LMs.

We use our method to detect errors in LM generation in the context of factual question-answering.

---

[1]In practice, EXAMINER and EXAMINEE can be the same LM (e.g., GPT-3) that is prompted in two different ways to define its different roles.

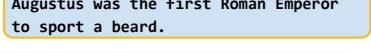

> **Augustus was the first Roman Emperor to sport a beard.**

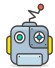

1. Who was Augustus?
2. What is the definition of a Roman Emperor?
3. Was it common for Roman Emperors to wear beards?
4. Who were the previous Roman Emperors before Augustus?

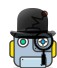

1. **Augustus was the first Roman Emperor.**
2. A Roman Emperor was a political leader who held supreme authority in the Roman Empire.
3. **No, it wasn't common for Roman Emperors to wear beards.**
4. The previous rulers before Augustus were known as Roman Republic's leaders, such as Julius Caesar

Are there any historical records or paintings that depict Augustus with a beard?

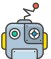

**There are no known historical records or paintings that depict Augustus with a beard.**

the claim that Augustus was the first Roman Emperor to sport a beard **is incorrect**. In fact, **it was not common for Roman Emperors to wear beards**, and Augustus was known for promoting a clean-shaven look. Additionally, **there are no known historical records or paintings that depict Augustus with a beard**.

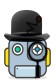

Figure 1: An example of our LMvLM approach. The first line shows the statement made by the EXAMINEE LLM. Then an interaction between the EXAMINER and EXAMINEE takes place, and the EXAMINER arrives at a conclusion whether the original statement was correct or not (here it concludes that it was a false statement).

Our experiments with several recent LMs ( CHAT-GPT, GPT-3 and LLAMA) show that cross-examination effectively detects factually incorrect claims generated by LMs. Specifically, across multiple datasets and examination settings, it detects over 70% of the incorrect claims while maintaining a high precision of >80%, outperforming strong baselines by a large gap.

Further analysis shows that examiner LMs introduce multiple questions throughout the examination, and employ various strategies to reveal inconsistencies, including question paraphrasing, validation of implicated arguments, claim decomposition, and requests for evidence.

To conclude, our contributions are (a) framing the task of factuality testing as an interaction between two LMs, (b) proposing a concrete implementation of this interaction via the use of one LM with different prompts in a zero-shot setting, and

(c) demonstrating improved factuality detection accuracy across several benchmarks.

## 2   LM Cross-Examination

Our goal is to employ an "examiner" LM (EXAMINER) to evaluate claims generated by another LM (EXAMINEE). To this end, we leverage the recent success of prompting (Liu et al., 2023b), to facilitate a cross-examination setting between the two LMs. In such a setting, EXAMINER should introduce questions with the objective of revealing inconsistencies with respect to an initial claim made by EXAMINEE. Such inconsistencies can be considered as a signal for uncertainty of EXAMINEE in its original claim, and thus can be used to assess whether its original statement was correct.

Given an EXAMINER LM and a claim $C$ generated by an EXAMINEE, our method establishes a multi-turn interaction between the LMs, where at each turn the other LM is prompted with a designated prompt that incorporates the outputs from previous turns. This interaction continues until the examiner has no further questions and can provide its final decision. To establish a meaningful interaction that reveals possible inconsistencies, we define three stages for the examination, each guided by a specific prompt. As part of each prompt for EXAMINEE or EXAMINER, we provide the outputs generated in the previous rounds for context. We next describe the examination stages in detail, with the overall process illustrated in Figure 2.

**Stage 1: Setup**   The examination begins by "assigning" the EXAMINER its role. Namely, describing the task setting, providing it with the EXAMINEE's claim, and asking it to generate questions for the EXAMINEE.[2]

Next, we feed the questions generated by EXAMINER, one at a time, to EXAMINEE, concatenated to the following instructions: `Please answer the following questions regarding your claim.` The response from EXAMINEE yields a set of answers to the questions from EXAMINER.

**Stage 2: Follow-up Questions**   We next feed EXAMINER with the answers generated by EXAMINEE to its initial questions, and ask EXAMINER whether it has any follow-up questions. Notably, outputs from EXAMINER at this stage are conditioned on

---

[2]We observe that this effectively steers EXAMINER to ask natural questions directly related to the given claim $C$ (§5).

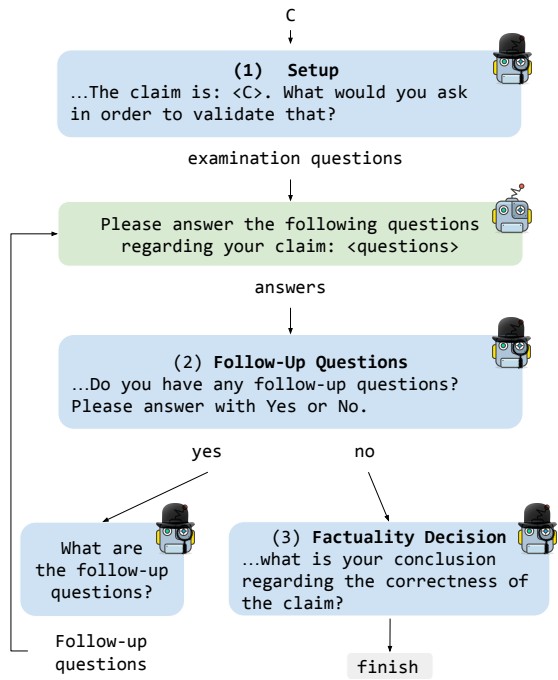

Figure 2: The three-stage process of cross-examination between the EXAMINER and EXAMINEE, where the factuality of a claim $C$ generated by EXAMINEE is estimated by EXAMINER.

the previous output from EXAMINEE. If the answer from EXAMINER is "Yes", we then further prompt it to obtain more questions. This phase is conducted iteratively, until either EXAMINER declares it has no follow-up questions, or the number of turns has reached a threshold.[3]

**Stage 3: Factuality Decision** Once no further questions are obtained from EXAMINER, we prompt it to conclude whether the claim $C$ is true or false. Specifically, we request it to reply with either "correct" or "incorrect" as its final conclusion. In cases where the examiner does not output either of "correct" or "incorrect", we consider its final decision to be a rejection of the claim.[4] Typically though, we observe that the examiner follows the instructions and indeed generates a definitive conclusion (see statistics in §5).

## 3 Related Work

**Attribution and Fact Checking** Our goal is closely related to works that check if LM-generated texts are faithful to a given source text (Bohnet et al., 2022; Honovich et al., 2022). This problem has been addressed via several approaches,

including question generation (Wang et al., 2020; Honovich et al., 2021; Scialom et al., 2021), NLI (Thorne et al., 2018; Welleck et al., 2019; Maynez et al., 2020; Dziri et al., 2022; Gao et al., 2022; Kamoi et al., 2023), data augmentation (Atanasova et al., 2022; Wright et al., 2022; Gekhman et al., 2023), and planning schemes that allow the model to self-edit its own generation (Schick et al., 2022). Unlike these works, we are not assuming any reference text or external knowledge bases. Instead, we directly check if the LM's claim is likely to be correct, by probing the model for inconsistencies. Our approach also uses multi-turn dialogue as a key component.

**Model Calibration** A key challenge with prediction models is to provide a probability of the answer being incorrect, a problem known as model calibration (Guo et al., 2017). The problem of factual-error detection can be viewed as a variation of calibration, where instead of a continuous probability, we provide a binary prediction for whether the model is correct or not. This is also related to the setting of selective prediction, where models can abstain from answering a query (Varshney et al., 2022; Kamath et al., 2020). Common approaches to calibration are to perform various transformations on model logits (Desai and Durrett, 2020; Jiang et al., 2021), and measuring uncertainty (e.g., see Kuhn et al., 2023). More recent works have studied the use of LMs for providing calibration, by training them on statements known to be factually correct or incorrect. This "supervised" approach has been explored via fine-tuning (Kadavath et al., 2022; Lin et al., 2022) and in-context learning (Cohen et al., 2023; Alivanistos et al., 2022).

We focus on zero-shot factual error detection that involves two categories: predicting whether a model's claim is correct or incorrect. While we focus on a binary setting, one could envision extensions of our approach to continuous outputs (i.e., the probability that the claim is correct).

**Multi-Agent LMs** Using multiple LMs in an interactive manner is a relatively new idea with many potential applications. It has been shown that LMs can utilize additional LMs or tools to enhance safety or better solve downstream tasks (Amodei et al., 2016; Irving et al., 2018; Barnes and Christiano, 2020; Schick et al., 2023). Additionally, Park et al. (2022) showed that in a social setting, LMs demonstrate certain social skills that emerge from

---

[3]We use a maximum of five turns in our experiments.

[4]We also search for additional indicative strings such as "true", "false", "is not correct", "is not true".

this interaction, and Shinn et al. (2023) proposes that an LM can use a different model to instruct it when to "reflect" on its recent action, while performing a planned sequence of actions aimed at solving a given query. Intuitively, this model detects signs of hallucination or inefficient planning within the LM's trajectory.

**Consistency Across Generations**  LMs have been shown to generate inconsistent outputs given different prompt paraphrases (Elazar et al., 2021; Newman et al., 2021). Prior work showed that prompts can be automatically optimized to produce factually correct claims more robustly (Lester et al., 2021; Zhong et al., 2021; Qin and Eisner, 2021). Hao et al. (2022) utilized multiple generated paraphrases to gauge consistency (Hao et al., 2022), and other works (Elazar et al., 2021; Zhou et al., 2022) further proposed training objectives to improve model consistency. Another approach to handling multiple outputs is via variants of decoding strategies (Wang et al., 2022), or model ensembles (Sun et al., 2022). In our work, we build on these, assuming inconsistencies are more likely to occur with incorrect claims, and let an examiner model search for these by introducing questions to the examinee.

**Chain of Thought Reasoning**  Recent work has shown that LMs can be prompted to elaborate on their reasoning process and that this could be exploited to improve mathematical, multi-hop and common-sense reasoning skills (Wei et al., 2022; Press et al., 2022; Yoran et al., 2023), along with planning and problem-solving abilities (Huang et al., 2022; Long, 2023). Another interesting approach to complex reasoning in LMs is recent work on Maieutic prompting (Jung et al., 2022), that answers a question by recursively generating a set of facts and reasoning over those.

Our approach may be viewed as constructing an elaborate chain-of-thought explanation for the examinee's claim. However, we do not train this explanation via in-context or fine-tuning, and rather rely on different prompts for its generation.

## 4  Experiments

In this section, we conduct experiments on multiple datasets and models to evaluate our approach, focusing on the task of factual question-answering.

| EXAMINEE | LAMA | TriviaQA | NQ | PopQA |
|---|---|---|---|---|
| LLAMA-7B | 53.9 | 48.4 | 33.8 | 24.9 |
| GPT-3 | 79.8 | 74.2 | 50.1 | 43.9 |
| CHATGPT | 80.9 | 77.2 | 53.3 | 45.6 |

Table 1: Portion of factually correct claims by every EXAMINEE LM on each dataset.

### 4.1  Experimental Setup

**Factual Question Answering**  One key use-case of LMs is answering questions seeking factual knowledge. For example, *"How old was Barack Obama when he was first elected?"*. In such cases, it is crucial for the model to answer the question correctly, or to indicate that it does not know the answer. We thus evaluate our approach on several Question Answering and Fact Completion datasets. These are typically provided as a set of $(Q, A)$ pairs of a question $Q$ and its ground-truth answer $A$. Having gold answers allows us to evaluate if a predicted answer is factually correct or not, which can be used to evaluate our LMvLM approach.

To apply cross-examination in this setting, we first convert the answer predicted by the model into an EXAMINEE claim that can be provided as input to the examination procedure. Formally, given a question $Q$, if $Q$ is phrased as a fill-in-the-blank question (e.g. *"Bailey Peninsula is located in ____"*), then we feed it to the EXAMINEE model to obtain a prediction that completes the sentence and forms a claim. In cases where $Q$ is phrased as a question (e.g., *"Where is Bailey Peninsula located?"*), we prompt the model to provide an answer in a claim format with: "Please answer the following question: `<Q>` Please phrase your answer as a claim.". This process results in a claim $C$ that states the model's "belief" about the answer to $Q$. We then evaluate the truthfulness of $C$ through cross-examination, and compare the examiner's decision of whether $C$ is correct or not to the ground-truth correctness.

**Factuality Evaluation Labels**  To evaluate our method, it is necessary to have "gold decisions" to compare the examiner's decisions against. Such labels can be obtained from the ground-truth answers in the data. Namely, the decision for a claim $C$ is correct if it matches an evaluation of $C$ against the gold answer $A$. To evaluate if the claim $C$ obtained for a question $Q$ is correct with respect to the ground-truth answer $A$, we first check if $A$ or any of its aliases (if provided as part of the dataset, e.g., "FC Tottenham" and "Tottenham Hotspur")

appears as a sub-string in $C$ (Schick et al., 2023; Meng et al., 2022). Next, to avoid incorrect labels resulting from this automatic evaluation (Bulian et al., 2022), we manually review all the claims marked as incorrect in the first step, and fix any labeling mistakes. We also filter out any ambiguous or unclear claims generated by EXAMINEE.

**Examiner Evaluation** We evaluate how well the examiner detects claims that are factually incorrect, using the following metrics:[5]

- **Precision**: the portion of incorrect claims, out of the claims rejected by the examiner.
- **Recall**: the portion of incorrect claims rejected by the examiner, out of all the incorrect claims.
- **F1**: the harmonic mean of precision and recall.

For completeness, we additionally report (in §E) the complementary Precision, Recall, and F1 scores with respect to detection of correct claims.

**Data** We consider the following datasets: LAMA (Petroni et al., 2019), TriviaQA (Joshi et al., 2017), Natural Questions (NQ) (Kwiatkowski et al., 2019) and PopQA (Mallen et al., 2022). These cover a wide range of queries, from real user queries (NQ), to trivia questions (TriviaQA), and subject-relation-object facts phrased as queries (LAMA, PopQA). We consider the closed-book open-ended setting, where we do not provide any context or answer choices to the model. We evaluate our approach on 1,000 random examples from the test set (or from the development set if a test set is not available).[6]

In addition, we created a dataset of false claims to further test our approach. This "Falsehoods" dataset contains only wrong claims, created separately for each model (GPT-3 and CHATGPT) and for each of the four QA datasets. Concretely, given a model and a question $Q$, we prompt the model to generate a false answer (see §C for details). We verify that these are indeed incorrect claims by checking that the gold answer (and any of its aliases, if they exist) does not occur in the generated text. This yields a subset of examples that are realistic, namely, the answer matches the target type (e.g., "a city") but is incorrect (see examples in Table 2). The examiner's decision for these examples should always be to reject.

---

[5]We say that the examiner "rejects" a claim if the examiner concludes that the claim is incorrect.

[6]We use only a subset of examples due to the high cost of running large LMs like GPT-3 used in our experiments.

| | False claim | Correct claim |
|---|---|---|
| GPT-3 | *"Wanlockhead is the highest village in France because it is located in the French Alps."* | *"Wanlockhead is the highest village in Scotland, a country in Europe."* |
| | *"Louis Oosthuizen the 2010 Open Golf Champion is American, because he was born in the United States."* | *"Louis Oosthuizen, the 2010 Open Golf Champion, is South African."* |
| CHATGPT | *"The screenwriter for "Smile" was definitely Steven Spielberg."* | *"The screenwriter for "Smile" was Jerry Belson."* |
| | *"Fontenay is located in the beautiful country of Antarctica."* | *"Fontenay is located in France."* |

Table 2: Example false claims generated by CHATGPT for PopQA and by GPT-3 for TriviaQA.

**Models** We use CHATGPT (`gpt-3.5-turbo`), GPT-3 (`text-davinci-003`) (Brown et al., 2020; Ouyang et al., 2022), and LLAMA-7B (Touvron et al., 2023), in three EXAMINER vs. EXAMINEE cross-examination settings: GPT-3 vs. GPT-3, CHATGPT vs. CHATGPT, and CHATGPT vs. LLAMA. Notably, using the same LM as EXAMINER and EXAMINEE (except for their prompts, which are different), provides a cleaner setting where both LMs share the same knowledge. The prompts used for each LM at every stage of the examination are shown in Table 10.

**Baselines** For each setting, we compare LMvLM with recent methods for uncertainty detection and variants of our approach:

- **Confidence-Based**: The prediction head of LMs outputs a probability for the predicted token. It is a common practice to use this probability as a measure of confidence in the prediction (Yoshikawa and Okazaki, 2023). In our case, the LM generates a multi-token claim, and we calculate the confidence for the claim as the product of probabilities for all predicted tokens of the answer only. In order to output a binary decision (i.e., is the claim correct or not), we optimize a threshold over the train dataset to maximize F1. Note that our examination approach does not require tuning any threshold.
- **Are you sure? (AYS)**: Recent work (Kadavath et al., 2022; Cohen et al., 2023) has shown that LMs can be trained to estimate their certainty in generated facts. Here, we use a zero-shot version of this approach where we directly "ask" the model whether it is sure. Specifically, we add the following prompt right after the claim generation:

| | LAMA | | | TriviaQA | | | NQ | | | PopQA | | |
|---|---|---|---|---|---|---|---|---|---|---|---|---|
| | **P** | **R** | **F1** | **P** | **R** | **F1** | **P** | **R** | **F1** | **P** | **R** | **F1** |
| AYS | 82.3 | 25.2 | 38.6 | 79.9 | 17.9 | 29.2 | 85.2 | 29.1 | 43.3 | 78.4 | 35.7 | 63.9 |
| IDK | 49.1 | 52.4 | 50.7 | 48.7 | 66.5 | 56.2 | 62.5 | 60.7 | 61.6 | 70.0 | 61.1 | 65.2 |
| **LMvLM** | 85.1 | 70.7 | 76.7 | 82.8 | 71.6 | 76.8 | 74.5 | 74.9 | 77.7 | 83.6 | 77.1 | 80.2 |
| **LMvLM** (Majority) | **86.6** | **75.8** | **80.8** | **84.5** | **80.8** | **82.6** | **82.3** | **76.1** | **79.1** | **87.0** | **84.0** | **85.4** |
| - Follow-up | 83.8 | 68.1 | 75.1 | 82.3 | 69.7 | 75.5 | 74.8 | 72.1 | 73.4 | 82.0 | 73.3 | 77.4 |

Table 3: Precision (P), Recall (R), and F1 scores for LMvLM with CHATGPT as EXAMINER and EXAMINEE, compared to baselines. The last row shows an ablation of our method without the follow-up questions stage.

| | LAMA | | | TriviaQA | | | NQ | | | PopQA | | |
|---|---|---|---|---|---|---|---|---|---|---|---|---|
| | **P** | **R** | **F1** | **P** | **R** | **F1** | **P** | **R** | **F1** | **P** | **R** | **F1** |
| AYS | 74.8 | 17.9 | 28.9 | 80.3 | 19.8 | 31.8 | 74.9 | 20.7 | 32.3 | 74.6 | 22.7 | 34.8 |
| IDK | 43.0 | 42.1 | 42.5 | 47.9 | 45.7 | 46.7 | 60.9 | 45.3 | 52.0 | 52.1 | 37.6 | 43.7 |
| Confidence-Based | 38.6 | **85.8** | 53.2 | 39.6 | **84.4** | 53.9 | 56.2 | 72.7 | 63.4 | 60.8 | 69.7 | 64.9 |
| IC-IDK | 71.5 | 46.3 | 56.2 | 70.6 | 49.7 | 60.1 | 70.0 | 57.6 | 63.2 | 76.9 | 37.7 | 50.6 |
| **LMvLM** | 78.8 | 69.9 | 74.1 | 81.6 | 64.6 | 72.1 | 70.5 | 66.6 | 68.5 | 75.5 | 69.1 | 72.2 |
| **LMvLM** (Majority) | **80.7** | 77.9 | **79.3** | **83.1** | 72.1 | **77.2** | **79.3** | **76.8** | **78.0** | **82.2** | **71.4** | **76.4** |
| - Follow-up | 76.4 | 71.1 | 73.7 | 78.7 | 64.8 | 71.1 | 66.6 | 70.1 | 68.3 | 70.9 | 65.8 | 68.3 |

Table 4: Precision (P), Recall (R), and F1 scores for LMvLM with GPT-3 as EXAMINER and EXAMINEE, compared to baselines. The last row shows an ablation of our method without the follow-up questions stage.

"Are you sure regarding the correctness of your claim? Please answer with Yes or No". Then we take the output as the prediction whether the claim is correct or not.

- **I don't know (IDK)**: Recently, Ganguli et al. (2023) showed that LMs might have the capability to self-correct themselves, when instructed to do so. Here we instruct the model to output *"I don't know"* if it is uncertain, by concatenating the following sentence to the original query: "If you are not sure you know the answer, answer with "I don't know" only.". If the model answers "I don't know" we label the corresponding claim as false, and otherwise true.

- **In-context IDK (IC-IDK)**: We teach the model to output that it doesn't know the answer, via in-context demonstrations. We follow Cohen et al. (2023) and test each of the queries within an in-context setting. For each query, we first provide the model with $K$ demonstrations, with $D$ of them labeled as *"Don't know"* examples, while the rest $K - D$ are provided with their gold answer from the dataset. The *"Don't know"* examples are randomly selected out of a set of examples the model failed on, while evaluating it on an held-out set of examples from the dataset in a zero-shot setting. Intuitively, these examples' answers are likely to be unknown to the model, hence we labeled them with *"Don't know"*. The model predictions are either a target

text or *"Don't know"*. Based on the output, we generate a factuality label as in the IDK baseline above. Notably, this baseline requires labeled data for the in-context demonstrations, which is not necessary for our approach.

- **LMvLM**: A single execution of our method, where we accept or reject the claim according to the examiner's final decision.

- **LMvLM (Majority)**: For a given claim, we apply our method three times (with the same EXAMINER and EXAMINEE), using sampling generation for follow-up questions generation. We reject the claim in case at least two of the examinations concluded it is false.

Since output probabilities are not provided as part of the CHATGPT's API, we cannot provide results for the Confidence-Based baselines in this case. Moreover, we observe that CHATGPT often fails to understand the task of IC-IDK.

### 4.2 Results

Tables 3, 4, 5 show the results for the the following pairs of EXAMINEE vs EXAMINER: CHATGPT vs. CHATGPT, GPT-3 vs. GPT-3, and LLAMA vs. CHATGPT, respectively. We do not include results for LLAMA as an EXAMINER since it did not work well, possibly because it is less specialized for instruction following.

Across all settings, our method outperforms the baselines, often by a large gap. For example, it ob-

| | LAMA | | | TriviaQA | | | NQ | | | PopQA | | |
|---|---|---|---|---|---|---|---|---|---|---|---|---|
| | **P** | **R** | **F1** | **P** | **R** | **F1** | **P** | **R** | **F1** | **P** | **R** | **F1** |
| AYS | 61.4 | 38.0 | 46.9 | 60.0 | 35.7 | 44.8 | 71.1 | 15.0 | 24.8 | 74.8 | 14.2 | 23.9 |
| IC-IDK | 56.6 | 49.0 | 52.5 | 58.9 | 52.5 | 55.5 | 66.2 | 53.4 | 59.1 | 66.8 | 50.1 | 57.3 |
| IDK | 61.6 | 44.8 | 51.9 | 62.0 | 32.9 | 43.0 | 64.4 | 12.1 | 20.4 | 66.7 | 16.8 | 26.8 |
| Confidence-Based | 54.9 | **76.7** | 64.0 | 56.9 | **85.8** | 68.4 | 64.4 | 63.5 | 63.9 | 64.6 | 53.6 | 58.6 |
| **LMvLM** | 81.1 | 66.4 | 73.0 | 80.1 | 70.8 | 75.2 | 79.3 | 65.5 | 71.7 | 84.9 | 73.6 | 78.8 |
| **LMvLM** (Majority) | **82.9** | 73.9 | **78.1** | **80.3** | 76.8 | **78.5** | **83.7** | **74.2** | **78.7** | **88.3** | **77.4** | **82.5** |
| - Follow-up | 79.7 | 65.7 | 72.0 | 80.0 | 69.8 | 74.6 | 79.4 | 63.7 | 70.7 | 83.3 | 71.8 | 77.1 |

Table 5: Precision (P), Recall (R), and F1 scores for LMvLM with CHATGPT as EXAMINER and LLAMA as EXAMINEE, compared to baselines. The last row is an ablation of our method without the follow-up questions stage.

| | LAMA | TriviaQA | NQ | PopQA |
|---|---|---|---|---|
| GPT-3 | 65.7 | 98.4 | 89.9 | 83.1 |
| GPT-3 (Majority) | 75.8 | 98.5 | 92.0 | 88.0 |
| CHATGPT | 83.6 | 97.9 | 90.4 | 88.8 |
| CHATGPT (Majority) | 87.1 | 98.6 | 94.2 | 93.9 |

Table 6: Accuracy of GPT-3 and CHATGPT as EXAMINER on false claims generated for each dataset.

| | CHATGPT / CHATGPT | GPT-3 / GPT-3 | CHATGPT / LLAMA |
|---|---|---|---|
| # of questions | 7.0 ± 2.8 | 6.4 ± 4.3 | 6.8 ± 4.4 |
| # of follow-up questions per iteration | 1.3 ± 1.0 | 1.3 ± 0.6 | 1.1 ± 0.5 |
| # of follow-up iterations | 1.9 ± 1.2 | 1.3 ± 0.7 | 1.6 ± 1.0 |
| # of questions per iteration | 3.1 ± 2.1 | 2.7 ± 1.6 | 2.9 ± 1.9 |
| % of inconclusive examiner decisions | 14.8% | 9.1% | 10.3% |

Table 7: Cross-examination statistics for each setting (EXAMINER/EXAMINEE), averaged across datasets.

tains 85.4 F1 compared to ≤ 65.2 by baselines for CHATGPT on PopQA (Table 3), and 77.2 F1 compared to ≤ 60.1 for GPT-3 on TriviaQA (Table 4). Notably, the most substantial gains are in terms of recall, showing the superiority of our method in detecting factually incorrect claims (when compared to the baselines which achieve reasonable precision too). Interestingly, we observe that CHATGPT generally outperforms GPT-3.

Last, Table 6 shows the accuracy of our method and baselines on our Falsehood dataset. For both CHATGPT and GPT-3, LMvLM successfully rejects a large majority of the false claims, obtaining 87%-98% accuracy with CHATGPT and 75%-99% with GPT-3 across all datasets.

## 4.3 Ablations

**Follow-Up Removal** we remove the follow-up iterations in the examination process to gauge their benefit. Results are reported for GPT-3 in Table 4 (last row), showing a large decrease in performance (e.g. 78 → 68.3 in F1 for NQ and 77.2 → 71.1 for TriviaQA). Notably, recall scores are decreased by 6%-10%. Overall, this shows the importance of the follow-up questions issued by the examiner to assess the examinee's claim.

**Retrieval-Augmented LMs** Language models can be significantly strengthened when augmented with additional retrieved data (Khandelwal et al., 2019; Borgeaud et al., 2022; Zhong et al., 2022; Guu et al., 2020). We next perform an evaluation of LMs in this setting as well. We focus on augmenting the EXAMINEE, since this is the model that can presumably benefit from additional information when answering questions. Specifically, we used DPR (Karpukhin et al., 2020) for retrieval and took the top three passages it retrieved from Wikipedia, concatenated them, and instructed a GPT-3 EXAMINEE (in the prompt) to answer the question based on the passages.

This resulted in improved accuracy of the EXAMINEE (i.e., the fraction of questions answered correctly) from 50.1 to 66.4 when augmented with DPR. Table 9 reports results on factuality detection when using GPT-3 as EXAMINER. It can be seen that our approach still outperforms the baselines, as in the case without augmentation.

Comparing the performance of the retrieval-augmented EXAMINEE, to the non-augmented one (see Table 4, NQ columns), we see that augmenting the EXAMINEE with retrieval leads to a substantial increase in precision (79.3 → 87.5) with only a small decrease in recall (76.8 → 75.7) and an overall improvement of 2.9% in F1. This shows that, alongside the improvement of the EXAMINEE,

| Pattern | Example statements/questions generated by EXAMINER during examination |
|---|---|
| Rephrasing the claim | Claim: *"The first Fast and Furious film was released in 2001."*
`In which year was the first Fast and Furious film released?` |
| Rephrasing Questions | Claim: *"The screenwriter who is credited with writing the screenplay for Winner is Wendy Riss"*
`1. What is the name of the screenwriter who is credited with writing the screenplay`
`for Winner? 2. Who is credited with writing the screenplay for Winner?` |
| Validation of Implications | Claim: *"The director of The Town was Ben Affleck."*
`Is Ben Affleck known for directing any movies?` |
| Logical decomposition | Claim: *"The second oldest of the Pevensie children in C S Lewis's The Lion, the Witch and the Wardrobe is Edmund."*
`1. What is the birth order of the Pevensie children in C S Lewis's The Lion, the`
`Witch and the Wardrobe? 2. What are their ages? 3. Who appears second in this list?` |
| Request for attribution | Claim: *"The screenwriter of Cover Up is Bill Blake"*
`Is there any evidence or documentation that supports the claim that Bill Blake was`
`the screenwriter for Cover Up?` |
| Wrong intermediate answers | Claim: *"There are eight vertices (corners) on an octahedron."*
`EXAMINER: How many vertices does an octahedron have? EXAMINEE: An octahedron has`
`eight vertices, each of which is the point where three edges meet.` |

Table 8: Examples for frequent patterns of CHATGPT and GPT-3 observed through manual analysis of cross-examinations.

|  | P | R | F1 |
|---|---|---|---|
| AYS | 80.2 | 20.1 | 32.1 |
| IDK | 69.5 | 46.0 | 55.4 |
| Confidence-Based | 62.6 | 71.9 | 66.9 |
| **LMvLM** | **87**.**5** | **75**.**7** | **80**.**3** |

Table 9: Precision (P), Recall (R), and F1 scores for LMvLM with GPT-3 as EXAMINER and a retrieval augmented GPT-3 as EXAMINEE.

LMvLM still performs well in detecting factual errors when they occur.

## 5 Analysis of Cross-Examinations

We analyze cross-examinations by GPT-3 and CHATGPT to better understand the success and failure cases of our method. We find that examiner LMs typically ask multiple questions in the examination, and perhaps surprisingly, seems to apply different strategies to reveal inconsistencies. We note that furhter analysis is required in order to better understand whether the EXAMINER indeed utilizes the conducted examination in its factuality decisions.

**Examination Statistics** Table 7 provides statistics on the cross-examinations performed by CHATGPT and GPT-3. Both models introduce multiple queries (6-7 on average) during an examination, with typically 1-2 steps of follow-up questions, which are important for the examiner's decision (§4.3). We also observe a non-negligible number

of claims (9%-15%) where the examiner LM does not arrive at a concrete final decision (i.e., it does not generate "correct" or "incorrect" as the final decision. We reject the claim in those cases). In our qualitative analysis, we identify reasons that could explain these cases.

**Qualitative Analysis** We manually analyze a sample of 96 examinations – 48 by each LM, with 6 correct and 6 incorrect examinations for each model and each dataset. We observe the following trends (examples are in Table 8):

1. **Rephrasing the claim**: In about 60% of the examinations, both LMs introduce questions which are paraphrases of the original question. This supports the assumption that the EXAMINER seeks inconsistencies by generating variants of the original claim.

2. **Rephrasing Questions**: In about half of the cases, both LMs introduce questions that are similar to previously asked questions or are differently phrased. This is a desirable behavior as it can reveal inconsistencies if the examinee provides a different answer for the same question.

3. **Validation of logical implications**: The EXAMINER asks EXAMINEE regarding implied arguments that must be true whenever the original claim is correct. This can be observed in 70% of the correct detections of GPT-3, and 87.5% out of the correct detections of CHATGPT.

4. **Logical questions**: The EXAMINER decomposes the claim into multiple sub-questions

which together compose a trajectory to validating it. Such decompositions appear in about 75% of the cases for CHATGPT but only 10% in GPT-3. We observe these in 33% of the correct detections of GPT-3, and 70% for CHATGPT.

5. **Request for attribution**: The EXAMINER ask the EXAMINEE about the existence of external evidence to support the claim. This happens in about 30% of the cases for both LMs.

6. **Wrong intermediate answers**: The EXAMINEE responds with factually incorrect answers to one or more of the questions asked by the EXAMINER. We observe this occurs mostly in cases where the original claim is false (it happens in only in about 14% of the cases where the EXAMINEE is correct). In both models, this can be observed in about half of the cases where the claim is false and has also been detected by the EXAMINER. Furthermore, it occurs in about 80% of the cases where the EXAMINER has accepted a false claim, and in 45% where the EXAMINER has rejected a correct claim.

We note that in most cases where LMvLM fails, EXAMINEE provides incorrect information to EXAMINER. This may indicate that in those cases EXAMINEE encodes a large set of factually wrong facts that are mutually consistent, thus making it hard for EXAMINER to detect inconsistencies. Finally, the fact that CHATGPT more commonly validates the claim via logical questions might be a key factor in its superiority over GPT-3 in our setting.

## 6 Conclusion

We introduce LMvLM, a method for zero-shot detection of factuality errors. Our method uses prompting to facilitate multi-turn interactions between an two LMs, to reveal inconsistencies that imply factually incorrect claims. LMvLM builds on a fundamental connection between self-consistency (i.e., consistency of an LM with itself) and factual consistency (i.e., consistency between claims generated by an LM and ground-truth facts). We consider the LM as the source of information, and we test whether a claim it has generated is consistent with other beliefs it has.

Our work can be extended in several ways. First, LMvLM provides interpretable information about related beliefs of the model, which could be analyzed to understand what makes the model commit certain mistakes. Second, one may incorporate several LM instances into the factuality detection process, rather the having only a single EXAMINER. Finally, one can train the EXAMINER to generate questions more effectively.

## Limitations

We note three limitations of our LMvLM method. First, it requires multiple queries of the examinee and examiner LMs, which could be costly when using external APIs such as those used in this work. This could be a key consideration when scaling this approach to large numbers of claims.

Second, for our method to succeed, both LMs (EXAMINEE and EXAMINER), but mostly EXAMINER, should be able to follow instructions and have the ability to reason over information in a relatively long context. This skill is currently mostly demonstrated by larger models (>10B parameters) and thus our method may not perform as well for smaller models.

Last, any logical flaws in the examiner's operation are likely to affect the overall examination, potentially leading to inaccurate decisions. However, our experiments show that, even if such flaws occur, our method is still useful on average as it substantially improves factuality detection. Nonetheless, developing safety mechanisms that detect and mitigate logical flaws is an important research direction, that we leave for future work.

## Acknowledgements

This work is supported in part by the Israeli Science Foundation. We thank Roee Aharoni and Avi Caciularu for valuable feedback and constructive suggestions.

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

## A Prompts

Table 10 provides the prompts used in our LMvLM approach.

## B Additional Evaluation

We follow the same experimental setting as in §4, but evaluate performance with respect to acceptance of claims rather than rejection. Specifically, here we use the following definitions:

- **Precision**: the portion of correct claims, out of the claims accepted by the examiner.
- **Recall**: the portion of correct claims accepted by the examiner, out of all the correct claims.

In addition, we introduce an ensemble **AYS + LMvLM**: for a given claim, we first run the AYS method, and if the claim is rejected by this method we then apply LMvLM (Majority) to obtain a final decision.

Tables 11 and 12 shows the evaluation results for the settings of CHATGPT vs. CHATGPT and GPT-3 vs. GPT-3, respectively.

In terms of precision, our method outperforms the other baselines, often by a large gap (e.g., $81.6$ compared to $\leq 60$ by baselines for CHATGPT on PopQA, and $68.7$ compared to $\leq 52.4$ for GPT-3 on PopQA). Moreover, this is while maintaining good recall with respect to the baselines, except for AYS which has the best recall scores.

## C Falsehoods Data

To generate a wrong claim, given a query $Q$ for one of the QA datasets we use, we prompt our models the following way: in case $Q$ is in a question format, we use "Please answer the following question with a wrong answer: $<Q>$" and further request the LM to "Please also phrase your answer as an argument". In case $Q$ is in a sentence-completion format, we use "Please complete the following sentence with a wrong answer: $<Q>$" and further concatenate $Q$ with the model answer. Table 13 introduces a few examples of these, generated by GPT-3. We manually verified that all the generated claims were indeed factually wrong.

## D Cross Examination Algorithm

Algorithm 1 provides psuedo-code for our method.

---

**Algorithm 1** Cross Examination

**Input:** A claim $C$ generated by EXAMINEE
**Output:** Correct / Incorrect

$Report \leftarrow$ ""
$Q \leftarrow$ EXAMINER$(\mathcal{P}_{setup}, C)$
$R_{curr} \leftarrow$ EXAMINEE$(C, Q)$
$Report \leftarrow Report + Q + R_{curr}$
**while** EXAMINER$(\mathcal{P}_{follow-up}, R)$ is "Yes" **do**
 $Q \leftarrow$ EXAMINER$(\mathcal{P}_{follow-up}, R)$
 $R_{prev} \leftarrow R_{curr}$
 $R_{curr} \leftarrow$ EXAMINEE$(R_{prev}, Q)$
 $Report \leftarrow Report + Q + R_{curr}$
**end while**
**return** EXAMINER$(\mathcal{P}_{decision}, C, Report)$

---

## E Example Cross-Examinations

Full cross-examination examples are provided in Tables 14, 15, 16, 17, 18, 19, 20, 21.

| Stage | GPT3 Prompt(s) | ChatGPT Prompt(s) |
|---|---|---|
| (1) Setup | Imagine trying to prove that a claim that someone claims is true, is wrong. You have the opportunity to ask any question in order to prove that the claim is: <C>. What would you ask in order to validate that? | Your goal is to try to verify the correctness of the following claim:<C>, based on the background information you will gather. To gather this, You will provide short questions whose purpose will be to verify the correctness of the claim, and I will reply to you with the answers to these. Hopefully, with the help of the background questions and their answers, you will be able to reach a conclusion as to whether the claim is correct or possibly incorrect. Please keep asking questions as long as you're yet to be sure regarding the true veracity of the claim. Please start with the first questions. |
| (2) Follow-Up Questions | (i) Do you have any follow-up questions? Please answer with Yes or No. 
 (ii) What are the follow-up questions? | |
| (3) Factuality Decision | Based on the interviewee's answers to your questions, what is your conclusion regarding the correctness of the claim? Do you think it is correct or incorrect? | |

Table 10: Prompts provided to EXAMINER at each stage of the examination, with respect to a claim $C$ by EXAMINEE.

| | LAMA | | | TriviaQA | | | NQ | | | PopQA | | |
|---|---|---|---|---|---|---|---|---|---|---|---|---|
| | **P** | **R** | **F1** | **P** | **R** | **F1** | **P** | **R** | **F1** | **P** | **R** | **F1** |
| IDK | 88.6 | 87.2 | 87.9 | 88.9 | 79.3 | 83.8 | 72.0 | 88.7 | 79.5 | 59.7 | 68.7 | 63.9 |
| AYS | 84.8 | 98.7 | 91.2 | 80.3 | 98.7 | 88.5 | 60.6 | 95.6 | 74.2 | 53.5 | 88.2 | 66.6 |
| **LMvLM** | 93.3 | 97.1 | 95.2 | 91.9 | 95.6 | 93.7 | 81.2 | 86.7 | 83.9 | 75.0 | 82.0 | 78.3 |
| **LMvLM** (Majority) | **94.5** | 97.2 | **95.8** | **94.4** | 95.6 | **95.0** | **85.1** | 88.4 | **86.7** | **81.7** | 85.0 | **83.3** |
| **AYS + LMvLM** (Ensemble) | 83.3 | **98.9** | 90.4 | 78.9 | **98.8** | 87.7 | 58.9 | **98.1** | 73.6 | 49.0 | **89.1** | 63.2 |

g

Table 11: Precision (P), Recall (R), both calculated with respect to acceptance, rather than rejection (see B), and F1 scores of CHATGPT as EXAMINER and EXAMINEE.

| | LAMA | | | TriviaQA | | | NQ | | | PopQA | | |
|---|---|---|---|---|---|---|---|---|---|---|---|---|
| | **P** | **R** | **F1** | **P** | **R** | **F1** | **P** | **R** | **F1** | **P** | **R** | **F1** |
| Confidence-Based | **94.8** | 65.4 | 77.4 | **91.1** | 55.2 | 68.7 | **84.8** | 30.2 | 44.5 | 52.4 | 42.6 | 47.0 |
| IDK | 85.4 | 85.9 | 85.6 | 81.4 | 82.7 | 82.1 | 56.6 | 71.0 | 63.0 | 41.2 | 44.9 | 47.4 |
| IC-IDK | 85.2 | 96.6 | 90.5 | 84.1 | 92.8 | 88.3 | 64.1 | 75.4 | 69.3 | 51.8 | 85.5 | 64.5 |
| AYS | 82.6 | 98.5 | 89.8 | 77.9 | 98.3 | 86.9 | 54.1 | 93.1 | 68.4 | 47.7 | 90.1 | 62.4 |
| **LMvLM** | 92.6 | 95.2 | 93.9 | 88.5 | 94.9 | 91.6 | 68.5 | 72.2 | 70.3 | 64.4 | 71.3 | 67.7 |
| **LMvLM** (Majority) | 94.5 | 95.3 | **94.9** | 90.4 | 94.9 | **92.6** | 77.6 | 80.0 | **78.8** | **68.7** | 80.2 | **74.0** |
| **AYS + LMvLM** (Ensemble) | 80.5 | **98.7** | 88.7 | 76.0 | **98.9** | 85.9 | 52.8 | **98.5** | 68.7 | 46.6 | **91.9** | 61.8 |

Table 12: Precision (P), Recall (R), both calculated with respect to acceptance, rather than rejection (see **??**), and F1 scores of GPT-3 as EXAMINER and EXAMINEE.

| False claim | True claim | Dataset |
|---|---|---|
| *"Windows is an operating system developed by Apple."* | *"Windows is an operating system developed by Microsoft."* | LAMA |
| *"The Hispaniolan lizard cuckoo (Coccyzus longirostris) is a species of cuckoo in the Cuculidae family.It is found in the Dominican Republic and Honduras."* | *"The Hispaniolan lizard cuckoo (Coccyzus longirostris) is a species of cuckoo in the Cuculidae family.It is found in the Dominican Republic and Haiti."* | LAMA |
| *"The first modern electric battery was demonstrated by Thomas Edison, an American inventor."* | *"The first modern electric battery was demonstrated by Alessandro Volta."* | TriviaQA |
| *"I believe that the actor who played Rockford's father, "Rocky," in the TV series, "The Rockford Files," was Tom Selleck."* | *"The actor who played Rockford's father, "Rocky," in the TV series, "The Rockford Files," was Noah Beery Jr."* | TriviaQA |
| *"The Taurus Mountains are located in the United States, specifically in the state of California."* | *"The Taurus Mountains are located in the southern Turkey"* | NQ |
| *"I heard that Taylor Swift is doing the 2018 Super Bowl Half Time Show."* | *"Justin Timberlake was the featured performer in the 2018 Super Bowl Half Time Show."* | NQ |
| *"Red Velvet is a type of cake"* | *"Red Velvet is a genre of music."* | PopQA |
| *"The author of The Jacket is J.K. Rowling, because she is the author of the Harry Potter series."* | *"The author of The Jacker is Andrew Clements."* | PopQA |

Table 13: Examples from our FALSEHOOD dataset and the corresponding correct claims.

EXAMINEE : The song 'I'm in the Mood for Dancing' was written by Irish songwriter and musician, Ben Findon.

EXAMINER :
1. Who is the artist that recorded the song "I'm in the Mood for Dancing"?
2. When was the song "I'm in the Mood for Dancing" first released?
3. Who are the listed songwriters for "I'm in the Mood for Dancing"?
4. What is the nationality of songwriter Ben Findon?
5. Are there any credible sources or references that credit Ben Findon as the writer of "I'm in the Mood for Dancing"?

EXAMINEE : As an AI language model, I can answer the following questions based on available data:
1. The song "I'm in the Mood for Dancing" was originally recorded by the Irish family band The Nolan Sisters, later known as The Nolans.
2. The song "I'm in the Mood for Dancing" was first released in 1979, the year in which it became a major international hit.
3. The listed songwriters for "I'm in the Mood for Dancing" are Ben Findon, Mike Myers, and Robert Puzey.
4. Ben Findon is a British songwriter.
5. Yes, there are credible sources that credit Ben Findon as the co-writer of "I'm in the Mood for Dancing." The song is registered with several music licensing agencies, such as ASCAP and BMI, which list Ben Findon as one of the song's writers. Additionally, the credits for the song are also listed in the liner notes of The Nolans' debut album of the same name.

EXAMINER : Can you provide any other sources or references that credit Ben Findon as the co-writer of "I'm in the Mood for Dancing"?

EXAMINEE : As an AI language model, I can provide additional sources or references that credit Ben Findon as the co-writer of "I'm in the Mood for Dancing."
1. AllMusic, a music database and online resource, lists Ben Findon as one of the songwriters for "I'm in the Mood for Dancing."
2. BMI, a performing rights organization, lists Ben Findon as a co-writer of "I'm in the Mood for Dancing" in their repertoire of musical works.
3. The official website of The Nolans also credits Ben Findon as one of the songwriters for "I'm in the Mood for Dancing."
4. The website of Discogs, a database and online marketplace for music, lists Ben Findon as a credited songwriter for the song.
5. The liner notes of The Nolans' album "I'm in the Mood..." credit Ben Findon, Mike Myers, and Robert Puzey as the song's writers.
These sources all provide additional evidence that Ben Findon is a co-writer of "I'm in the Mood for Dancing."

EXAMINER : Based on the available information, the claim that "The song 'I'm in the Mood for Dancing' was written by Irish songwriter and musician, Ben Findon" **is incorrect**. It appears that Ben Findon is only one of the co-writers of the song, alongside Mike Myers and Robert Puzey. The song was originally recorded by the Irish family band The Nolan Sisters, later known as The Nolans.

Table 14: Example of successful cross-examination with CHATGPT as EXAMINER and EXAMINEE, for the claim *"The song 'I'm in the Mood for Dancing' was written by Irish songwriter and musician, Ben Findon."* (which is only partly correct, and has been labeled as incorrect). For brevity, prompts are not shown.

```
EXAMINEE : The producer of Precious was Lee Daniels.

EXAMINER :
1. Can you provide a list of the producers involved in the production of Precious?
2. Who was the director of Precious?
3. Did Lee Daniels win any awards for producing Precious?
4. What other movies has Lee Daniels produced?
5.  Are there any interviews or articles where Lee Daniels specifically states that he was the
producer of Precious?

EXAMINEE :
1. Sure, here is a list of producers involved in the production of "Precious": Lee Daniels, Gary
Magness, Sarah Siegel-Magness, Oprah Winfrey, Tom Heller, and Tyler Perry.
2. The director of "Precious" is Lee Daniels.
3. Yes, Lee Daniels won several awards for producing "Precious," including the Academy Award for
Best Picture in 2010.
4. Lee Daniels has produced several other films throughout his career, including "The Paperboy,"
"Monster's Ball," "The Woodsman," and "The Butler."
5. Yes, there are several interviews and articles in which Lee Daniels specifically states that he
was one of the producers of "Precious." In fact, he is often credited as one of the driving forces
behind the film's success and is known for his efforts to bring the story to the screen.
EXAMINER : Based on the provided answers, the claim that the producer of "Precious" was Lee Daniels
is correct.
```

Table 15: Example of successful cross-examination with CHATGPT as EXAMINER and EXAMINEE, for the claim *"The producer of Precious was Lee Daniels."* (which is correct). For brevity, prompts are not shown.

```
EXAMINEE : There are eight vertices (corners) on an octahedron

EXAMINER :
1. What is an octahedron?
2. How many faces does an octahedron have?
3. How many edges does an octahedron have?
4. How many vertices does an octahedron have?
5. Are all vertices of an octahedron located at the same distance from the center?

EXAMINEE :
1. An octahedron is a three-dimensional geometric shape that is made up of eight equal equilateral
triangles. It is a polyhedron with eight faces.
2. An octahedron has eight faces, each of which is an equilateral triangle.
3. An octahedron has 12 edges that connect its eight vertices.
4. An octahedron has eight vertices, each of which is the point where three edges meet.
5. No, not all vertices of an octahedron are located at the same distance from the center. There are
two types of vertices in an octahedron: those at the corners of the shape, which are equidistant
from the center, and those located at the center of each face, which are further away from the
center than the corner vertices.
EXAMINER :  Based on the answers, the claim that "there are eight vertices on an octahedron" is
correct.
```

Table 16: Example of failed cross-examination with CHATGPT as EXAMINER and EXAMINEE, for the claim *"There are eight vertices (corners) on an octahedron."* (which is incorrect). For brevity, prompts are not shown.

```
EXAMINEE : Augustus was the first Roman Emperor to sport a beard.

EXAMINER :
1.Who was Augustus?
2. What is the definition of a Roman Emperor?
3. Was it common for Roman Emperors to wear beards?
4. Who were the previous Roman Emperors before Augustus?
5. Are there any historical records or paintings that depict Augustus with a beard?

EXAMINEE :
1. Augustus was the first Roman Emperor who ruled from 27 BCE to 14 CE. He was the successor of
Julius Caesar and the founder of the Roman Empire.
2. A Roman Emperor was a political leader who held supreme authority in the Roman Empire, including
military, judicial, and legislative powers. The position was established after Julius Caesar's
assassination, and Augustus was the first to hold the title officially.
3. No, it wasn't common for Roman Emperors to wear beards. In fact, most Roman Emperors were
clean-shaven, and a beard was perceived as a symbol of being barbaric or uncivilized.
4.The previous rulers before Augustus were known as Roman Republic's leaders, such as Julius Caesar,
Pompey the Great, Marcus Licinius Crassus, and others.
5. There are no known historical records or paintings that depict Augustus with a beard. Most of the
depictions show him with a clean-shaven face or a small mustache. Additionally, Augustus himself
was known for promoting a clean-shaven look, and he even encouraged his advisors and officials to
do the same.
EXAMINER : Based on the information that has been gathered, the claim that Augustus was the first
Roman Emperor to sport a beard is incorrect. In fact, it was not common for Roman Emperors to wear
beards, and Augustus was known for promoting a clean-shaven look. Additionally, there are no known
historical records or paintings that depict Augustus with a beard.
```

Table 17: Example of successful cross-examination with CHATGPT as EXAMINER and EXAMINEE, for the claim *"Augustus was the first Roman Emperor to sport a beard."* (which is incorrect). For brevity, prompts are not shown.

```
EXAMINEE : The capital of Eritrea is Asmara.

EXAMINER : What is the official record of the capital of Eritrea according to the government?

EXAMINEE : The official record of the capital of Eritrea according to the government is Asmara.

EXAMINER : Does the government of Eritrea recognize any other cities as official capitals?

EXAMINEE : No, the government of Eritrea only recognizes Asmara as its official capital.

EXAMINER : Based on the answers to my questions, I conclude that the claim that the capital of
Eritrea is Asmara is correct.
```

Table 18: Example of successful cross-examination with GPT-3 as EXAMINER and EXAMINEE, for the claim *"The capital of Eritrea is Asmara"* (which is correct). For brevity, prompts are not shown.

```
EXAMINEE : Croatian composer Igor Stravinsky was born in Zagreb.

EXAMINER : Where was Igor Stravinsky born?

EXAMINEE : Igor Stravinsky was born in Oranienbaum, Russia (now Lomonosov, Russia) on June 17,
1882.
EXAMINER : What was the nationality of Igor Stravinsky's parents?

EXAMINEE : Igor Stravinsky's parents were both Russian.

EXAMINER : Based on the information provided, the claim that Croatian composer Igor Stravinsky was
born in Zagreb is incorrect.
```

Table 19: Example of successful cross-examination with GPT-3 as EXAMINER and EXAMINEE, for the claim *"Croatian composer Igor Stravinsky was born in Zagreb"* (which is incorrect). For brevity, prompts are not shown.

```
EXAMINEE : Cristiano Ronaldo joined Manchester United in 2003.

EXAMINER : Which club did Cristiano Ronaldo join in 2003?

EXAMINEE : Cristiano Ronaldo joined Sporting CP in 2003.

EXAMINER : do you have any proof that Cristiano Ronaldo joined Sporting CP in 2003?

EXAMINEE : Yes, there is proof that Cristiano Ronaldo joined Sporting CP in 2003. Sporting CP has
released official documents confirming the transfer, and there are also numerous news articles and
interviews from the time that confirm the move.
EXAMINER : Based on the evidence provided, it is clear that the claim that Cristiano Ronaldo joined
Manchester United in 2003 is false. He actually joined Sporting CP in 2003.
```

Table 20: Example of failed cross-examination with GPT-3 as EXAMINER and EXAMINEE, for the claim *"Cristiano Ronaldo joined Manchester United in 2003"* (which is correct). For brevity, prompts are not shown.

```
EXAMINEE : The United States Capitol Building is topped by the Statue of Freedom.

EXAMINER : What is the actual figure that is located on top of the United States Capitol Building?

EXAMINEE :  The actual figure located on top of the United States Capitol Building is a bronze
statue of the Roman goddess Freedom, which stands 19 feet 6 inches tall.
EXAMINER :  Is the figure known as the Statue of Freedom?

EXAMINEE : No, the figure is not known as the Statue of Freedom. It is known as the Statue of
Liberty.
EXAMINER : Based on the answers to the questions, the claim that the United States Capitol Building
is topped by the Statue of Freedom is incorrect. The actual figure located on top of the United
States Capitol Building is a bronze statue of the Roman goddess Freedom, which is not known as the
Statue of Freedom.
```

Table 21: Example of failed cross-examination with GPT-3 as EXAMINER and EXAMINEE, for the claim *"The United States Capitol Building is topped by the Statue of Freedom"* (which is correct). For brevity, prompts are not shown.