# OpenReview forum: "LM vs LM: Detecting Factual Errors via Cross Examination"
_EMNLP/2023/Conference — EMNLP 2023 Main_

### Official Review · Reviewer_ZiT3 · 2023-08-03

**Soundness:** 4

**Excitement:**

4: Strong: This paper deepens the understanding of some phenomenon or lowers the barriers to an existing research direction.

**Paper Topic And Main Contributions:**

The paper presents the LMvLM framework, which is a method to assess factual errors in open-domain, closed-book QA answers of LLMs. In the framework, which resembles "cross-examination" in the legal domain, an "Examiner" repeatedly questions the "Examinee" with regard to their claim, and finally decides whether they believe the original answer of the model is accurate/inaccurate. The paper experiments with multiple generative QA datasets, and three LLM-pair settings (GPT3-GPT3), (CGPT-CGPT) and (Llama-CGPT). The results convincingly find that this conversational method of assessing accuracy of an answer is more performant than baselines that include asking the Examinee for its assurance, or using LM probability with a threshold. Analysis reveals that successful Examiners employ diverse strategies and often do multiple rounds of questioning.

**Questions For The Authors:**

- I personally did not find the experiment on false synthetic claims (Table 7) to be particularly interesting. It seems likely that the LLM would generate blatantly false claims (which is confirmed in the examples of Table 2), which leads to very high performance on this set. It is not clear why this experiment is needed in the paper. What does this experiment reveal that the other main experiment on real error cases does not?
- One major question I have is: is the # questions asked correlated with a successful outcome? If so, one could imagine artificially "forcing" an LLM to generate more questions (by making it believe it decided to ask more questions), and measure whether this would lead to real gains in measure.
- Do you have thoughts on applying the protocol on other task settings that involve factuality checking, such as summarization, KG-grounded generation, retrieval-enabled QA, etc.?
- It seems like the ultimate objective should be to obtain a Examinee that produces better answers (higher numbers in Table 1). Can the framework accomodate for the Examinee to modify its answer?

**Reasons To Accept:**

- A interesting & innovative approach to factuality verification in the context of closed-book QA, which is interpretable.
- Broad experiments on 4 datasets, confirming the generalizability of the experiments.
- The paper is convincing on the quality of the data, with the authors carefully describing their annotation procedure (for example regarding gold label assignment, or final assignment of the Examiner score), which adds evidence that the benchmark resource could be valuable to other researchers if released.

**Reasons To Reject:**

Nothing major, but here are some notes:
- It seems limiting that the authors only included experiments with three LM-pairs. There are several interesting questions to explore that could yield insights into how to apply the procedure most successfully. Namely: does the Examiner need to be more "proficient" at the initial task for this to be successful, or are less "powerful" Examiners still valuable in uncovering errors? How does the "performance gap" between a model pair impact performance. How does the performance of the Examiner on each particular sample impact its ability to be a good examiner? I find it unfortunate that 2/3 pairs included involve the same LM for both roles, it seems limiting.
- Closed-book but open-domain QA seems somewhat synthetic. Nowadays, retrieval capabilities are mainstream. It seems likely that some of the models included in the experiments would be much less likely to make mistakes given retrieved evidence, rendering the experimental setup less valuable. Perhaps a Discussion section addressing this, and giving thoughts on how to adapt the framework to a setting with retrieval would be valuable.

**Reproducibility:**

5: Could easily reproduce the results.

**Reviewer Confidence:**

4: Quite sure. I tried to check the important points carefully. It's unlikely, though conceivable, that I missed something that should affect my ratings.

**Typos Grammar Style And Presentation Improvements:**

- Regarding the name. The title uses "LM vs LM" and most of the paper mentions "LMvLM", which both seem a bit generic. I particularly enjoyed the "cross examination" analogy, why not go for something more specific and eye-catching such as "CrossExam"?

---

> ### Author Rebuttal · Authors · 2023-08-29
>
> We thank the reviewer for their thorough review and thoughtful comments. We are encouraged that the reviewer finds our approach to be interesting and innovative, and the experiments to be broad.
>
> ### Response to Weakness 1:
> It’s indeed a very interesting question to explore different models and their interactions. We focused on three “popular” models, but indeed others are also interesting. We note that the “Examiner LM” needs to be able to understand and follow instructions, for successfully conducting the full “Cross-Examination” (this is why we didn’t use LLama as an examiner). We will add a discussion of these interesting questions in the camera-ready version.
>
> ### Response to Weakness 2:
> Thanks for this valuable suggestion! Indeed it is possible to also consider retrieval augmented QA. Technically this would be simple to do by replacing the examinee and/or the examiner with a retrieval augmented model. For the examiner, we don’t expect results in this case to be qualitatively different from regular QA. Namely, we expect the examiner to still probe the examinee with questions that seek inconsistencies, as with “context-less QA”.
> Nonetheless, we agree that evaluating a stronger examinee that is augmented with retrieval is interesting and important. Therefore, we followed your response and re-evaluated “GPT3 vs GPT3” on NQ, but now using a retrieval-augmented examinee – We used DPR for retrieval and took the top 3 passages it retrieved from Wikipedia, concatenated them, and instructed the examinee (in the prompt) to answer the question based on the passages.
>
> In terms of the examinee performance, we note a substantial improvement in accuracy (i.e., the fraction of questions answered correctly) from 50.1 to 66.4 when augmented with DPR.
> **For the performance of our method compared to the baselines, we observe similar trends to the “context-less QA” case**
> (see Table 4 in the submission, columns for NQ):
>
> | **Method** | **Precision** | **Recall** |  **F1** |
> | -------- | ------- | ------- | ------- |
> | AYS (+retrieval augmented examinee) | 80.2 | 20.1 | 32.1 |
> | IDK (+retrieval augmented examinee) | 69.5 | 46.0 | 55.4 |
> | Confidence-Based (+retrieval augmented examinee) | 62.6 | 71.9 | 66.9 |
> | LMvsLM (examinee without retrieval)  | **79.3** | **76.8** | **78.0** |
> | LMvsLM (+retrieval-augmented examinee)  | **87.5** | **75.7** | **80.3** |
>
> Comparing the last two rows in the table, we see that augmenting the examinee with retrieval leads to a substantial increase in precision (79.3 → 87.5) with only a small decrease in recall (76.8 → 75.7) and an overall improvement of 2.9% in F1.
> **This shows that, alongside the improvement of the examinee, “LM vs LM” still performs well in detecting factual errors when they occur.**
> We will provide more extensive experiments and discussion about that in the camera ready version.
>
>
> We thank the reviewer for the great and insightful questions, which we address below.
> ### Answer to Question 1:
> The  “Falsehood” dataset is meant to be a controlled test for false-claim detection, which we think is useful as a general “sanity-check” in this setting, and useful to future works on factuality. We agree that could be made more challenging, however, there’s still a room for improvement as for example GPT3 (as an Examiner) fails on 25%-35% of the claims generated based on LAMA. Yet, we will modify the camera ready version to contain a “hard-negatives” version of this dataset. This can be done in various ways, including training a separate classifier to detect falsehoods, and filter out examples detected by the classifier. Thank you for this comment!
>
>
> ### Answer to Question 2:
> Indeed we have observed (Section  4.3) that using follow-up questions helps the “Examiner LM'' achieve better results. Following your suggestion, we calculated the correlation coefficient between the success of the “Examiner LM” to the number of questions it asks, and obtained a value of 0.43. This indeed suggests a dependence between these variables. Consequently, we have conducted the following experiment: we randomly sampled 200 cases (50 from each of the four datasets we use) where the examination fails, and for each of the cases we did not allow the examination to end before it gets to at least 8 ‘follow-up questions’ iterations. Indeed, for 10% of the cases the factuality decision of the examiner had switched (that is, became to be correct). Therefore, in the camera-ready version we will report on additional ways of making the examiner ask more questions.
>
> ### Answer to Question 3:
> Thanks again for the great question! Our approach can indeed be adapted to other tasks where factual statements may be generated. Yet, automatically evaluating the approach on those settings may be not trivial as well, and requires additional manual effort to annotate the correctness of the “Examinee LM” generations. We will definitely add a discussion of this in the camera-ready version.
>
> ### Answer to Question 4:
> Great thought again! We agree that ultimately we would like to get the “Examinee LM” to produce more accurate answers. The point is it is reasonable to think that in some cases where the “Examinee LM” fails, it is really lacking the relevant parametric knowledge. One idea could be to use retrieval after applying our approach, in cases when the “Examiner LM” concludes that the claim is wrong, and thus to reduce the calls to the retriever. We will add a discussion of this direction in the camera-ready version.

---

### Official Review · Reviewer_eV78 · 2023-08-04

**Soundness:** 3

**Excitement:**

3: Ambivalent: It has merits (e.g., it reports state-of-the-art results, the idea is nice), but there are key weaknesses (e.g., it describes incremental work), and it can significantly benefit from another round of revision. However, I won't object to accepting it if my co-reviewers champion it.

**Paper Topic And Main Contributions:**

This paper's contributions are as follows:
- framing the task of factuality testing as an interaction between two LMs,
- proposing a concrete implementation of this interaction via the use of one LM with different prompts in a zero-shot setting, and
- demonstrating improved factuality detection accuracy across several benchmarks.

**Reasons To Accept:**

This paper confirms the effectiveness of the proposed method in the experiment.

**Reasons To Reject:**

Although the basic concept of this paper is very reasonable and I agree with it, I am concerned that the evaluation using a set of QA is an evaluation that trivializes the proposed method.

**Reproducibility:**

3: Could reproduce the results with some difficulty. The settings of parameters are underspecified or subjectively determined; the training/evaluation data are not widely available.

**Reviewer Confidence:**

2: Willing to defend my evaluation, but it is fairly likely that I missed some details, didn't understand some central points, or can't be sure about the novelty of the work.

---

> ### Author Rebuttal · Authors · 2023-08-29
>
> First, thanks for the time reviewing our work.
> We respectfully disagree with the claim that “using a set of QA is an evaluation that trivializes the proposed method”. In fact, answering questions is a key capability of LLMs, and many of its uses are precisely for this task (e.g., a large fraction of search queries are questions. We also use such queries here via the NaturalQuestions dataset). And, within this important task of QA factuality is still an unsolved problem, which significantly hinders the use of LLMs. Thus, we think QA is a very appropriate and challenging platform for working on the problem of factuality.
> We agree that there are other tasks where factuality can be evaluated (e.g., summarization, conversation, fact-checking), but we don’t see why these are better than the QA setting we use here.
>
> Here are a few more works which use QA as a format for evaluation: https://aclanthology.org/2021.emnlp-main.619/, https://aclanthology.org/2020.findings-emnlp.117/, https://doi.org/10.18653/v1/2022.findings-acl.158, https://arxiv.org/abs/2207.05221, https://arxiv.org/abs/2205.14334, https://arxiv.org/abs/2208.11057, https://arxiv.org/pdf/2302.09664.pdf, https://aclanthology.org/2023.findings-eacl.150/.

---

### Official Review · Reviewer_hr8G · 2023-08-04

**Soundness:** 3

**Excitement:**

4: Strong: This paper deepens the understanding of some phenomenon or lowers the barriers to an existing research direction.

**Paper Topic And Main Contributions:**

The paper proposes a method to verify claims made by an LLM (presumably as answer to a question) through a process of cross-examination: one LLM (possibly a different instance of the same LLM) generates questions about the claim under consideration, and through a multi-turn process reaches a conclusion whether the original claim can be trusted.

The paper includes a good analysis of previous work that puts the effort in context:

* Attribution and fact checking: check whether generated text is faithful to a given source text. Difference: the authors do not assume explicit source text, but test self-consistency.

* Model calibration: prob that prediction is true / turned into categorical decision (statement is true).  Could extend work to calibration with continuous prob.

* Multi-agent LMs: "social skills" - would like to see instead a discussion of why different LMs (with different parameters) would improve each other.

* Consistency across generations - self-consistency models.

* COT: improves mathematical, multi-hop, commonsense reasoning abilities.  The proposed approach can be thought as Chain of Thought generation of an explanation for a claim - but without ICL or fine-tuning.

Experiments assess the performance of this cross-examination method on multiple datasets (LAMA / triviaQA / NQ / PopQA).

Multiple baselines are considered:

1. Confidence based: product of prob of each token (372).  (The Independence assumption must be checked.)

2. Are you Sure (AYS): "LLMs can be trained to estimate their certainty" (Kadavath et al 2023, Cohen et al 2023) - but this is not trained here - zero-shot "AYS".

3. IDK - add in prompt option to answer "IDK"  - Ganguli 2023

4. IC-IDK - ICL for IDK prediction - trained on generated falsehoods

Tests: LMvLM (1 run) and LMvLM (3 runs - majority)

Results are reported in three configurations:
* ChatGPT vs ChatGPT
* GPT3 vs GPT3
* ChatGPT vs Llama 7b

Table 3 shows the best results when using ChatGPT for cross-examination - and obtaining factuality judgment with highest F-measure compared to baselines.  For models that provide access to token-confidence, the confidence baseline is the best (Table 5) but cross-examination still performs better.

A qualitative analysis of the cross-examination process identifies "examination strategies" that are employed by the examiner LLM: rephrasing claim / question / validation of implication / logical decomposition / request for attribution / wrong intermediate answers.







**Questions For The Authors:**

A. It seems the authors specifically aim at designing a method that does not require any training - of any form: pre-training, fine-tuning, ICL - on either the task itself or on any of the auxiliary sub-tasks (like falsehood generation).  Why do you take this a priori decision? I understand the benefit of using a "black box" LLM "out of the box" without any additional work - but at least for baselines, which trained approaches could you consider as comparison points?  (The baselines you compare to are not trained - even the "Are you Sure" methods which are trained in previous work are used here without training).

B. Is there a basis to believe that LLMs are globally consistent across generations wrt a specific claim? How would you assess this claim?

C. You generate a set of "falsehoods" as part of the process - do you aim for these falsehoods to be hard to detect? How can we measure the adversarial nature of your synthesized data? How can you increase its adversarial level?

D. Section 5 identifies cross-examination strategies in observed data: Table 8 - different methods used by examiner LLMs:
rephrasing claim / question / validation of implication / logical decomposition / request for attribution / wrong intermediate answers.

These are indeed good strategies - but isn't this a form of "confirmation bias"? A baseline would consist of constructively pursue these specific methods for validation. Why would we rely on the LLM to come up with "validation strategies" instead of instructing it to use these well known methods - either through ICL or through RLHF.


**Reasons To Accept:**

A. The paper addresses what is probably the most acute issue with current LLM technology - the risk of generating unsupported non-factual claims.  It provides an effective technique to detect such false claims that requires no training or fine-tuning.

B. Experimental analysis is thorough - across multiple datasets and baselines and shows high performance.

C. Analysis of previous work is illuminating - it offers a useful synthesis of different methods and trends grouped in informative manner.

**Reasons To Reject:**

A. The motivation behind the approach is that LLMs have a tendency to produce globally consistent claims across multiple turns (072-076).
There is, however, no basis to hold this assumption as true.  [Actually - work of Eleazar et al 2021 aims at encouraging such consistency through fine-tuning]

B. 087-092 - unclear what is the point of this paragraph: "examiner LMs introduce multiple questions throughout the examination, and employ various strategies to reveal inconsistencies, including question paraphrasing, validation of implicated arguments, claim decomposition, and requests for evidence."
Is it "emerging behavior" attributed to the "examiner LMs" or is it designed by the authors through clever prompting?
I feel uneasy attributing such "strategic planning" capabilities to LLMs without any task-specific training.
The observations reported in Section 5 are not sufficient ground to establish the presence of such "strategies".

"this complex reasoning is naturally performed via the strong conversational skills of modern LMs"
I am worried about associating "complex reasoning" skills with conversation - especially when conversation includes ungrounded / non-factual claims (it is the whole point of the paper to detect these).

C. The survey of previous work on "Consistency across Generations" is very well done - and it concludes with the statement: "we build on this... assuming inconsistencies are more likely to occur with incorrect claims, and let an examiner model search for these by introducing questions to the examinee."  But actually - this statement is NOT building on previous work which demonstrated that LLMs do NOT have this property of triggering more inconsistencies with incorrect claims.

I would instead expect "building upon previous work" to include an experiment that measures empirically specifically whether self-consistency is more prevalent when starting from a false claim or a true claim, and whether this prevalence can be improved through intervention (training, specific prompting, ICL, fine-tuning).

D. 338-352 and Appendix C describe how the dataset of "falsehoods" is generated.  It sounds like the aim is to generate "adversarial falsehoods" (that is, falsehoods that are difficult to distinguish from true claims).  But the examples in Table 2 are not at all adversarial.  It seems that the process should be validated in a more robust way - by creating a falsehood classifier using simple training.








**Reproducibility:**

4: Could mostly reproduce the results, but there may be some variation because of sample variance or minor variations in their interpretation of the protocol or method.

**Reviewer Confidence:**

5: Positive that my evaluation is correct. I read the paper very carefully and I am very familiar with related work.

---

> ### Author Rebuttal · Authors · 2023-08-29
>
> We thank the reviewer for their thorough review and thoughtful comments. We are encouraged that the reviewer finds our method to be effective, the experimental analysis thorough and the analysis of previous work illuminating.
>
> We address the reviewer’s concerns raised under “reasons to reject”:
> ### Response to Concern 1
> Indeed, the main incentive has been to propose a method which requires as little effort as possible, by avoiding any kind of training or dependency on annotated training examples.
> Additionally, in order to conduct a fair comparison with the other baselines, we have used most of them without additional training, except for the IC-IDK method which indeed utilizes ICL. We also note that many models of interest (e.g., ChatGPT) do not allow fine-tuning, and we also wanted to study those (we understand that ICL could still be an option for there).
> However, we definitely agree that this method (and the other baselines) might have benefited from additional training, including RLHF, but decided to leave it for future work.
>
> ### Response to Concern 2:
> The purpose of this evaluation was mostly to achieve a high-level categorization of the  “Examiner LM'' generated text. We came up with the categories based on initial exploration of conversations, and then quantified their prevalence. Perhaps “strategies'' is a too strong  term, and we can use a different wording. We do think that the fact the model generates such conversations without explicit instruction is yet another testament to the complex skills that LLMs exhibit. Finally, the reviewer suggests (Q4) using prompting to induce specific interrogation behavior and this would indeed be very interesting to explore.
>
> ### Response to Concern 3:
> Indeed it would be interesting to further quantify the phenomenon of inconsistency in factual knowledge. We note that Elazar et al. 2021 mostly evaluates consistencies with respect to different paraphrases, and here our concern is with more elaborate forms of consistency.
> But, we note that if inconsistency (of the type LMvLM evaluated) had been the same for factually correct and incorrect facts, then our method would not have been able to improve performance on factuality testing (because true and false facts would have resulted in the same result). We also note that the results of Elazar were obtained for BERT, RoBERTa and ALBERT, and it is likely that models have since improved. We do agree it would be interesting to define several notions of inconsistency (e.g., negation, aliases, logical relations etc) and evaluate how LLMs behave on these.
>
> ### Response to Concern 4:
> Our goal with falsehoods data was actually not to generate adversarial examples, but rather to generate facts we know to be false, and evaluate LMvLM on these. Some of these examples may in fact be “easy” to detect. Note that even for these non-adversarial examples, we do see up to 34.3% of the cases reported as true, so there is headroom even for this simpler case.
> We however definitely agree with the reviewer that using hard negatives could produce a more challenging evaluation approach. One way to generate those would indeed be to use a trained classifier, or alternatively take cases where generation with temperature results in conflicting results. Following your suggestion, we will also report results on hard-negatives in the camera-ready. Thanks for this note!
>
>
> We answer the reviewer’s questions raised under “Questions For The Authors”:
> ### Answer to Question 1:
> Please see our response to concern 1 above.
>
> ### Answer to Question 2:
> Indeed our motivation for devising the method is that it is harder to answer consistently when one is lying, because it requires weaving a set of facts around the lie that is self consistent. This is why asking multiple questions is a way to detect lies in humans, and our expectation was that something similar would happen with LLMs, and our results indeed support this. The reviewer asks: “Is there a basis to believe that LLMs are globally consistent across generations”. Our results suggest that for facts that are true, LLMs provide more consistent answers, at least when consistency is measured by another LLM. It would be interesting to evaluate consistency itself more robustly by having a human provide the final “verdict” as to whether generation was consistent or not. We note that there is a recent line of work that evaluates LLMs on several “logically” related facts and sees if they produce consistent results (https://aclanthology.org/2023.findings-acl.733, http://arxiv.org/abs/2305.14956, http://arxiv.org/abs/2305.14795, https://arxiv.org/abs/2307.12976). It will be interesting to combine these directions with LMvLM.
>
> ### Answer to Question 3:
> Please see our response to concern 4 above.
>
> ### Answer to Question 4:
> Please see our response to concern 2 above.

---

### Meta-Review · Area_Chair_CQGn · 2023-09-15

**Recommendation:** 5

**Metareview:**

This paper proposes a method for better calibration in question answering / fact probing through cross examination: where they use a series of paraphrased questions / verification questions to the LM in order to judge whether the model’s answer is correct or not. Results demonstrate consistent improvements over 4 datasets.

The reviewers acknowledged that the problem tackled in the paper is a critical problem (hr8G), experiments and analysis are insightful  (hr8G, ZiT3) and effectively demonstrate proposed method is effective (eV78, ZiT3), and description of prior work is comprehensive (hr8G, ZiT3).

Some concerns raised by reviewers include: (1) “False” claims used in the experiments are too synthetic/simple (h48G), (2) evaluation in QA is too simple (eV78), and (3) lack of more LM pairs and discussion on the choice of the pair (ZiT3). Lack of experiments with retrieval was also pointed out (ZiT3) but was resolved in author responses. The authors are encouraged to add these additional experiments in the paper and add clarification in the experimental setting in the final version of the paper if accepted.

---

### Decision · Program_Chairs · 2023-10-07

**Decision:**

Accept-Main

**Comment:**

This paper proposes a method for better calibration in question answering / fact probing through cross examination: where they use a series of paraphrased questions / verification questions to the LM in order to judge whether the model’s answer is correct or not. Results demonstrate consistent improvements over 4 datasets.

The reviewers acknowledged that the problem tackled in the paper is a critical problem (hr8G), experiments and analysis are insightful  (hr8G, ZiT3) and effectively demonstrate proposed method is effective (eV78, ZiT3), and description of prior work is comprehensive (hr8G, ZiT3).

Some concerns raised by reviewers include: (1) “False” claims used in the experiments are too synthetic/simple (h48G), (2) evaluation in QA is too simple (eV78), and (3) lack of more LM pairs and discussion on the choice of the pair (ZiT3). Lack of experiments with retrieval was also pointed out (ZiT3) but was resolved in author responses. The authors are encouraged to add these additional experiments in the paper and add clarification in the experimental setting in the final version of the paper if accepted.